# Skeletal Muscle Fiber Adaptations Following Resistance Training Using Repetition Maximums or Relative Intensity

**DOI:** 10.3390/sports7070169

**Published:** 2019-07-11

**Authors:** Kevin M. Carroll, Caleb D. Bazyler, Jake R. Bernards, Christopher B. Taber, Charles A. Stuart, Brad H. DeWeese, Kimitake Sato, Michael H. Stone

**Affiliations:** 1Department of Sport, Exercise, Recreation, and Kinesiology, East Tennessee State University, Johnson City, TN 37614, USA; 2Department of Exercise Science, College of Health Professions, Sacred Heart University, Fairfield, CT 06825, USA; 3Department of Internal Medicine, Quillen College of Medicine, East Tennessee State University, Johnson City, TN 37614, USA

**Keywords:** muscle fiber, hypertrophy, muscle force, strength, athlete

## Abstract

The purpose of the study was to compare the physiological responses of skeletal muscle to a resistance training (RT) program using repetition maximum (RM) or relative intensity (RI_SR_). Fifteen well-trained males underwent RT 3 d·wk^−1^ for 10 weeks in either an RM group (*n* = 8) or RI_SR_ group (*n* = 7). The RM group achieved a relative maximum each day, while the RI_SR_ group trained based on percentages. The RM group exercised until muscular failure on each exercise, while the RI_SR_ group did not reach muscular failure throughout the intervention. Percutaneous needle biopsies of the vastus lateralis were obtained pre-post the training intervention, along with ultrasonography measures. Dependent variables were: Fiber type-specific cross-sectional area (CSA); anatomical CSA (ACSA); muscle thickness (MT); mammalian target of rapamycin (mTOR); adenosine monophosphate protein kinase (AMPK); and myosin heavy chains (MHC) specific for type I (MHC1), type IIA (MHC2A), and type IIX (MHC2X). Mixed-design analysis of variance and effect size using Hedge’s *g* were used to assess within- and between-group alterations. RI_SR_ statistically increased type I CSA (*p* = 0.018, *g* = 0.56), type II CSA (*p* = 0.012, *g* = 0.81), ACSA (*p* = 0.002, *g* = 0.53), and MT (*p* < 0.001, *g* = 1.47). RI_SR_ also yielded a significant mTOR reduction (*p* = 0.031, *g* = −1.40). Conversely, RM statistically increased only MT (*p* = 0.003, *g* = 0.80). Between-group effect sizes supported RI_SR_ for type I CSA (*g* = 0.48), type II CSA (*g* = 0.50), ACSA (*g* = 1.03), MT (*g* = 0.72), MHC2X (*g* = 0.31), MHC2A (*g* = 0.87), and MHC1 (*g* = 0.59); with all other effects being of trivial magnitude (*g* < 0.20). Our results demonstrated greater adaptations in fiber size, whole-muscle size, and several key contractile proteins when using RI_SR_ compared to RM loading paradigms.

## 1. Introduction

Recent evidence has suggested performance outcomes favored resistance training (RT) using relative intensity (RI_SR_), where training is not performed to muscular failure, compared to repetition maximum (RM) training, where training to muscular failure is achieved during each resistance exercise [1]. It was hypothesized that these preferential benefits of RI_SR_ training were, in part, due to superior fatigue management through the use of heavy-and-light training days and non-failure training sessions used throughout the intervention. Conversely, RM training consisted of very high relative intensity (i.e., failure) training every session with little variability, possibly impacting the group’s ability to recover and adapt appropriately [2]. Performance outcomes, such as maximal force production, rate of force development, and vertical jump performance [1], are certainly critical in understanding any training program’s efficacy. However, a more thorough investigation of the underlying mechanisms within the skeletal muscle tissue is warranted.

Sarcomeres, the functional units of skeletal muscle, are central contributors to the activity and capability of the muscle. Alterations in protein isoforms within the sarcomere give rise to skeletal muscle plasticity or changes in phenotype. Myosin heavy-chain (MHC) isoforms are directly related to the muscle fiber type [3,4] and the shortening velocity of the fiber [5,6]. Alterations and synthesis of MHC isoforms provide a great deal of information regarding training outcomes. Further, the addition of more sarcomeres and the MHCs that they contain is the basis for muscle hypertrophy [7]. Because of their degree of involvement in contraction dynamics, these factors are often considered when examining training outcomes or comparing training programs [4,8,9]. Additionally, myofibrillar protein synthesis is, in part, controlled by a complex network of cellular signaling pathways [10,11]. Much of the divergence in myofibrillar vs. mitochondrial protein synthesis has been attributed to the interaction between the protein kinase B (PKB, or Akt)-mammalian target of rapamycin (mTOR) pathway and the adenosine monophosphate kinase (AMPK)-peroxisome proliferator-activated receptor gamma coactivator 1-alpha (PGC1α) pathway [12,13]. Activation of the Akt-mTOR pathway has been shown to increase following RT and plays a key role in the synthesis of myofibrillar proteins (such as MHC isoforms), while the Akt-mTOR pathway is inhibited via AMPK-PGC1α activation of the tuberous sclerosis complex 2 (TSC2) [14]. Therefore, when examining responses to training in the skeletal muscle, it seems prudent to perform at minimum a cursory analysis of these relevant signaling proteins.

Due to differences in load prescription (e.g., failure vs. non-failure), RI_SR_ and RM training may result in divergent cellular signaling responses, which could affect adaptations to the skeletal muscle tissue and ultimately performance. Thus, the purpose of the study was to compare the physiological responses of skeletal muscle between a RM or RI_SR_ resistance training program. We hypothesized that RI_SR_ would result in superior gains in muscle size and contractile protein content due to a more polarized model of training intensity allowing for greater fatigue management.

## 2. Materials and Methods

### 2.1. Subjects

Eighteen well-trained males volunteered for the study, however, one subject withdrew before beginning the training intervention and two others (one from each group) withdrew due to minor injuries during the study. As a result of these drop outs, 15 subjects (age = 26.94 ± 3.95 years, body mass = 86.21 ± 12.07 kg, BMI = 27.07 ± 3.08) completed the training intervention and were included in this analysis. Subject experience with consistent resistance training of at least a 3 days·wk^−1^ frequency was 7.7 ± 4.2 years, as confirmed by an exercise history questionnaire and careful questioning by the investigators. In addition to training history, we considered our subjects to be well trained based on their baseline isometric mid-thigh pull peak force (IPF) (4403.61 ± 664.69 N) and allometrically scaled isometric peak force (IPFa) (226.04 ± 25.81 N/k^−0.67^). These values are in line with previously published data in well-trained, competitive athletes [15,16,17]. The study groups were equated and formed by matching for baseline IPFa and assignment to either a RI_SR_ group using % set-rep best (RI_SR_, *n* = 7) or an RM zone group (RM, *n* = 8) where the final set of each exercise was taken to momentary muscular failure. It should be noted that group equalization was performed with the initial 18 subjects, prior to any dropouts. However, an independent samples *t*-test confirmed that there were no baseline differences between the groups (*p* > 0.05). All subjects read and signed an informed consent document prior to participating in the study, as approved by the university’s Institutional Review Board (ethical code: 0716.13f).

### 2.2. Resistance Training

Training methodology for the current study was extensively outlined in a previously published manuscript [1]. Both training groups completed resistance training 3 d·wk^−1^ for 10 weeks on Mondays, Wednesdays, and Fridays (Table 1 and Table 2). Additionally, sprint training was conducted 2 d·wk^−1^ throughout the intervention on Tuesdays and Thursdays and was identical for both groups.

Both group programs were based on a block-periodization approach [18,19,20], with the only difference between training groups being the loading strategy used. The RI_SR_ group used submaximal intensities (i.e., percentages) to guide the training process, while the RM group used maximal loads within each training session with the set and repetition prescription. Loads were adjusted for the RI_SR_ group based on estimated set-rep bests within each set-rep combination (e.g., 3 × 10, 3 × 5). Additionally, the RI_SR_ group used heavy and light days consistent with previous research and block-periodization concepts [18,19]. Conversely, the RM group adjusted loads based on the maximal load lifted in each training session, within the RM zone prescription (e.g., 3 × 8−12, 3 × 4−6). The estimated training volume load was equalized between groups by having each group’s set-and-repetition prescription aligned with one another (e.g., a 3 × 4−6 RM in the RM group was mirrored with a 3 × 5 in the RI_SR_ group) [1]. The RM zone training necessitated that each subject would reach muscular failure on the final set of the prescription, indicating a maximum effort had been achieved. These daily maximums were then used to adjust training loads for the subsequent session. If the failed set resulted in fewer repetitions than were prescribed in the RM zone, the load was reduced by a minimum of 2.5% for the next training session. However, if the repetitions achieved on the failed set exceeded the prescription, the load would be increased by a minimum of 2.5%. All other training factors not pertaining to the loading strategy were controlled to the best of our ability (e.g., coaching, training time).

Both groups performed the same dynamic warm-up preceding each training session and performed the same lift-specific warm-up procedures during resistance training. Specifically, each subject performed three progressive sets of warm-ups for each of the major lifts (squats, pulls, and presses). Maximum effort was encouraged on every set of every exercise throughout the intervention. Subjects were highly motivated and completed 100% of the prescribed training. Subjects were instructed to refrain from excess physical activity outside of training and on rest days. Lastly, every training session was closely supervised by multiple certified strength and conditioning coaches throughout the intervention.

### 2.3. Muscle Biopsy Sampling and Processing

Muscle biopsies were sampled at rest at least 72 h before any study activity and 72 h after the final training session. Following an overnight fast, a percutaneous needle biopsy of the vastus lateralis (VL) was obtained using a 5 mm Bergstrom-Stille biopsy needle under suction [21,22] and local anesthetic (1% lidocaine). The specimen was obtained in the superficial region of the VL at a depth of approximately 3 cm for both pre- and post-testing using the double-sample technique. Additionally, care was taken to obtain the post-sample at a distance 0.5 cm distal of the pre-sample and at the same tissue depth. After removal of fascia and other tissue, about half of the 50–100 mg sample was mounted on cork, quickly frozen in isopentane, and cooled in liquid nitrogen for later sectioning on a cryostat (Leica, Wetzlar, Germany) and immunohistochemical analysis. The remainder of the sample was placed in a container and frozen in an isopentane slurry cooled over liquid nitrogen. All samples were then promptly stored at −80 °C until they were needed for analysis.

The cork-mounted biopsy samples were removed from the −80 °C freezer and allowed to thaw to −20 °C. Serial sections were obtained of each sample at a thickness of 14 µm and affixed to a microscope slide. Following this, tissues were fixed with acetone at −20 °C for five minutes. All samples were blocked for two hours in a 10% normal goat serum and incubated overnight in monoclonal antibodies specific to myosin heavy-chain (MCH) isoforms: MHC2X for type IIX fibers (IgM, 1:10 dilution), MHC2A for type IIA fibers (IgG1, 1:100 dilution), and MHC1 for type I fibers (IgG2b, 1:200 dilution). Each of these antibodies were obtained from the Developmental Studies Hybridoma Bank (DSHB, University of Iowa, Iowa, USA). The following day, sections were incubated for two hours using goat anti-mouse AlexaFluor 488 (IgM), AlexaFluor 350 (IgG1), and AlexaFluor 555 (IgG2b), each at 1:200 dilution (Invitrogen, Carlsbad, CA, USA). A series of photographs were taken of the slides at 10× magnification using an Olympus BX41 microscope and imaged using an Olympus Qcolor3 camera. Images were processed in the ImageJ software (National Institute of Health, USA). A total of 3018 fibers were measured using the software’s tracing tool (100.6 fibers/sample on average), and the average circularity of the measured fibers was 0.77 ± 0.09. Fiber types were identified and sized based on the staining color within each fiber (Figure 1). Of the 30 biopsy samples (pre-and-post), only 13 of them were positive for type IIX muscle fibers (of those 13, only five had greater than 10 type IIX fibers). Therefore, type IIX and type IIA fiber sizes were not separated for statistical analyses.

Prior to immunoblot processing, a small piece of tissue was removed from −80 °C storage and kept on dry ice. Muscle homogenates were prepared by separating 25–50 mg of muscle into a solution consisting of 500 µL 0.25 M sucrose, 20 mM HEPES buffer, and protease inhibitors (Halt Protease Inhibitor Cocktail Kit; Pierce, Rockford, IL, USA). This solution was then homogenized with 2–3 15-s bursts of a homogenizer (Pellet Pestle Motor; Kontes, Vineland, NJ) as previously described [23]. Antibodies raised against mTOR and AMPK were purchased from Cell Signaling (Danvers, MA, USA), while MHC2X and MHC1 were purchased from Sigma Aldrich (St. Louid, MO, USA). Antibodies for MHC2A were obtained from the DSHB as mentioned above. For mTOR and AMPK analysis, samples containing 10 µg of protein was applied to 3% to 8% polyacrylamide gradient gels for immunoblotting, while 5 µg of protein was used for MHC2X, MHC2A, and MHC1. Following one hour of electrophoresis at 150 V, each gel was transferred to a polyvinylidene difluoride membrane. This transfer was performed for 90 min at 80 V. Each immunoblot was blocked in 5% nonfat dry milk for two hours prior to overnight incubation in the primary antibody. The following day, appropriate secondary antibodies were used at 1:5000 dilution for two hours prior to chemiluminescent imaging using a Syngene G:Box iChemi XT. Each of the samples were run in duplicate and the pre- and post-samples for each subject were run on the same gel (Figure 2). The odd numbered lanes on each gel contained the pre-samples, while the next even numbered lane contained the post samples for each respective subject.

### 2.4. Ultrasonography

Anatomical cross-sectional area (ACSA) and muscle thickness (MT) of the right leg, mid-vastus lateralis (VL) was assessed using ultrasonography (LOGIQ P6, General Electric Healthcare, Wauwatosa, WI, USA) on each subject before and after the intervention. Ultrasonography was performed 48–72 h following the most recent training. Prior to measurement, each subject’s hydration status was determined using refractometry (Atago, Tokyo, Japan) to ensure level of hydration would not affect the ultrasonography measures. Each subject began the ultrasonography session by lying on their left side with an internal knee angle of 170 ± 5°. To determine measurement site, landmarks were found and marked at the greater trochanter and lateral epicondyle of the femur. The length between these landmarks was the femur length, and 50% of this length was marked and used as the measurement site. Additionally, another marking was placed 5 cm medial to the 50% femur mark for MT measurement. The athlete’s femur length was recorded and used for subsequent testing sessions to ensure proper placement of the probe. Additionally, probe placement and orientation were verified by comparing adipose and connective tissue markings from previous images to the current image.

Following application of a water-soluble transmission gel, a 16 Hz ultrasonography probe was oriented perpendicular to the VL at 50% femur length. ACSA images were obtained using a panoramic sweep in the transverse plane of the VL using the LOGIQView function of the ultrasound device. For MT, the probe was oriented 5 cm medial to the mid-femur marking parallel with the VL. Utmost care was given to not depress the skin or tissues during measurement. Vastus lateralis ACSA was measured by tracing the inter-muscular interface in the cross-sectional images, and MT was measured as the distance between subcutaneous adipose tissue-muscle interface and inter-muscular interface. Three images were collected for each subject and were analyzed on the ultrasonography instrument. Nearly perfect reliability was observed using intraclass correlation coefficient (ACSA ICC = 0.99, CV = 1.75%; MT ICC = 1.00, CV = 0.77%); the three images were averaged together for statistical analysis.

### 2.5. Statistical Analysis

Data were assessed for normality using a Shapiro–Wilk test and for homogeneity of variance using a Levene’s test. To ensure that drop-outs did not affect baseline group differences, an independent samples *t*-test was performed and revealed no significant between-group differences at baseline (*p* > 0.05). A 2 × 2 (group × time) mixed design analysis of variance (ANOVA) was used to examine main effects for each of the variables derived from the muscle biopsy samples and ultrasonography. Statistically significant main effects were further examined using Holm’s sequential Bonferroni post-hoc adjustment. Effect size using Hedge’s *g* with 90% confidence intervals (CI) was calculated for each pre-post variable within-group and between-group effects to further examine the practical significance of these results. Effect size values of 0.0, 0.2, 0.6, 1.2, 2.0, and 4.0 were interpreted as trivial, small, moderate, large, very large, and nearly perfect, respectively [24]. The alpha level before post-hoc adjustments was set as *p* ≤ 0.05. Statistical analyses were performed on a commercially available statistics software (JASP version 0.8.1.1) and Microsoft Excel 2016 (Microsoft Corporation, Redmond, WA, USA).

## 3. Results

For measurement of muscle size, type I CSA, type II CSA, and MT each resulted in statistically significant main effects for time (*p* < 0.001), while there was a statistically significant interaction effect for ACSA (*p* = 0.046). There were no between-group differences at pre or post for ACSA; however, post-hoc tests revealed statistically significant increases for the RI_SR_ group in type I CSA (*p* = 0.018), type II CSA (*p* = 0.012), ACSA (*p* = 0.002), and MT (*p* < 0.001). With the exception of MT (*p* = 0.003), none of these measurements reached statistical significance for the RM group (*p* > 0.05) (Figure 3 and Figure 4). However, effect sizes for muscle size measurements revealed small-large effect sizes for the RI_SR_ group and small-moderate changes for the RM group. Between-group effect sizes favored the RI_SR_ group with small-moderate effect magnitudes (Table 3).

Basal levels of total mTOR decreased from pre to post, indicated by a statistically significant main effect for time (*p* = 0.007). Post-hoc tests revealed a statistically significant decrease in mTOR for the RI_SR_ group (*p* = 0.031) but not for the RM group (*p* = 0.08). No statistically significant main effects were observed for AMPK (*p* = 0.792), MHC2X (*p* = 0.072), MHC2A (*p* = 0.055), or MHC1 (*p* = 0.090) (Figure 5). Effect size statistics for the RI_SR_ group suggested a large decrease in total mTOR, trivial changes in total AMPK, and moderate increases for MHC2X, MHC2A, and MHC1. For the RM group, moderate decreases in mTOR were observed, no change in AMPK, and small increases in each of the myosin heavy chains. Between-group effect sizes again favored the RI_SR_ group for each of the myosin heavy chains with effect magnitudes ranging from small-moderate. mTOR and AMPK each had trivial between-group effects (Table 4).

## 4. Discussion

The main purpose of this study was to compare the skeletal muscle physiological alterations following either a relative intensity or repetition maximum program. In agreement with our hypothesis, the results of our investigation indicate that adaptations to whole muscle size, fiber size, and greater accretion of key myofibrillar proteins favored RI_SR_ training over RM training. Both groups trained using the same periodization scheme with no statistical differences in volume load [1], yet the results seemed to favor the RI_SR_ group. We propose that a major contributor to the result was superior fatigue management in the RI_SR_ group and that consistent training to failure in the RM group possibly led to a reduced ability to adapt in our well-trained sample.

Hypertrophic adaptations at both the whole muscle and single-fiber level favored the RI_SR_ over RM training group, evidenced by the small-to-moderate between-group effect magnitudes (*g* = 0.48–1.03). Higher volume loads have been associated with greater increases in muscle size [7]. Even using similar volume loads [1], the RI group resulted in greater size gains. Previous researchers have suggested training to failure permits the maximal recruitment of all motor units within a task [25], and thus provide optimal stimulation of both high- and low-threshold motor units regardless of training intensity. However, the results of the current investigation are contrary to this hypothesis, as RI_SR_ had greater magnitudes of type 2 and type 1 fiber size increases compared to the RM group. This is possibly due to a lack of recovery allowed by virtue of consistently training to failure in the RM group, rather than insufficient stimuli. In support of this, Moran-Navarro et al. [2] recently demonstrated that performing bench press and back squats to failure delays recovery of neuromuscular performance by up to 24–48 h post-exercise [2]. Further, the greater hypertrophy in the RI_SR_ group supports the use of a broader loading spectrum (e.g., heavy-and-light days, down sets) within a training week. Indeed, there is a paucity of data in well-trained individuals comparing the RI_SR_ and RM. Thus, to our knowledge, this study is the first to suggest that RI_SR_ yields more optimal adaptations compared to RM for muscle hypertrophy in strength-trained subjects.

Small-to-moderate between-group effect magnitudes supported the RI_SR_ group for MHC2X (*g* = 0.31), MHC2A (*g* = 0.87), MHC1 (*g* = 0.59). Although statistical significance (*p*-value) was not attained for any MHC isoform, the effect magnitudes support the RI_SR_ group. Indeed, the lack of statistical significance, particularly in the cases of MHC2X and MHC2A, may have been a result of small sample size. Future investigations including larger samples may reveal more information regarding these effects. The accretion of myofibrillar proteins is an important component of muscular performance [5,6]. The greater enhancements in MHC isoforms in the RI_SR_ group may provide information to why the RI_SR_ group also improved muscular performance more so than the RM group [1]. Conversely, the RM group’s lesser accretion of MHCs could be due to the increased fatigue and delayed recovery associated with RT to failure. Previous research has demonstrated failure training to induce greater levels of fatigue compared to non-failure training [2], which may impact the ability for meaningful accretion of myofibrillar proteins. MHC2X and MHC2A showed greater increases for both groups compared to MHC1, with the former being expressed in type IIX and type IIA muscle fibers, respectively. This suggests the RT stimulus, particularly in the RI_SR_ group, may have selectively enhanced production of faster isoforms of MHC. Although beyond the scope of the current study, tapering has been shown to produce an increase in fast MHC expression [26,27]. Thus, the taper performed by both groups during the last training phase may have influenced these alterations.

Alterations in total mTOR were somewhat small in magnitude compared to MHC (Figure 5). However, there was a large, statistically significant decrease in mTOR in the RI_SR_ group (*g* = −1.40), and a moderate, non-statistically significant decrease for the RM group (*g* = −0.97). Additionally, there were no significant changes in total AMPK levels in either group. The decreases in total mTOR are interesting, as most research examines mTOR alterations within an acute exercise window (i.e., 0–24 h post-exercise) and usually measures the level of mTOR (or its targets) activation [14,28,29,30]. Research examining the changes in basal total mTOR following RT interventions is sparse [23]. Additionally, acute mTOR increases are suppressed following repeated RT stimuli [31]. This suggests the decreases in basal mTOR in the current study may have been a result of a molecular adaptation. Additionally, there are various other, potentially mTOR-independent mechanisms by which protein translation may be initiated, such as via the costamere and focal adhesion kinase [32]. Although mTOR is a critical protein for cellular growth, it is also important to note that there are many interacting and competing signals within the in vivo environment of a skeletal muscle cell [33,34]. The combinations of these signals are likely the ultimate contributor to fiber and whole-muscle hypertrophy.

Several limitations within the study should be noted in an effort to bolster the interpretation and application of these data. First, muscle edema (i.e., swelling) was not accounted for in the ultrasound measurements of the VL. Previous research has suggested very early hypertrophic responses to training (~3 weeks) may be due to mostly muscle edema [35]. While this investigation did not control for edema, ACSA measurements were collected at least 48–72 h post exercise, and the post measurements were taken following a 10-week training program. Additionally, nutritional intake was not controlled between groups. Nutrient availability plays a role in the acute activation of intracellular signaling proteins mTOR and AMPK [36]; therefore, readers should exercise caution when interpreting the results of the chronic changes in signaling proteins observed in this investigation. However, it should be noted that changes in body mass across the study were not different between groups (*p* > 0.05) [1]. This indicates caloric intake did not differ greatly between training groups.

## 5. Conclusions

These results demonstrated a greater effect for fiber and whole-muscle CSA following RI_SR_ compared to RM training in well-trained males. Along with the increased muscle hypertrophy, the RI_SR_ group increased the content of several key MHC isoforms to a greater extent than the RM group, which may be explained by the greater variation in workload distribution in the RI_SR_ group through the use of heavy and light training and non-failure training sessions. These results, taken together with the previously published performance data [1], support the use of RI_SR_ training in well-trained populations over that of RM training.

## Figures and Tables

**Figure 1 sports-07-00169-f001:**
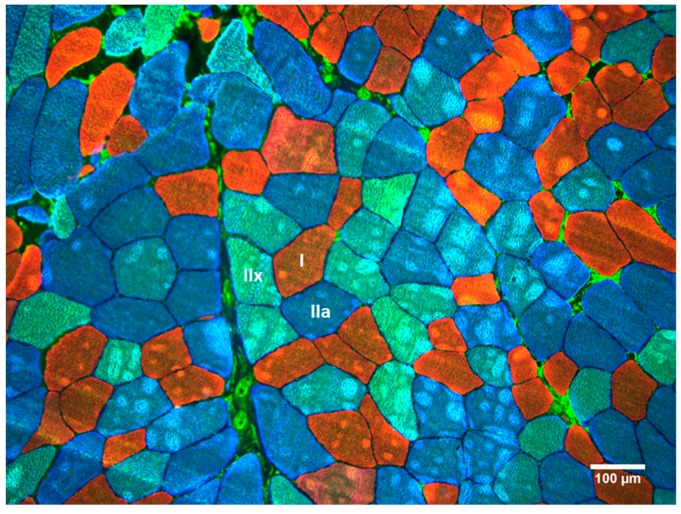
Example of histochemical stains for myosin heavy-chain (MHC) isoforms: MHC2X (type IIX; green), MHC2A (type IIA; blue), and MHC1 (type I; red). Scale = 100 µm.

**Figure 2 sports-07-00169-f002:**
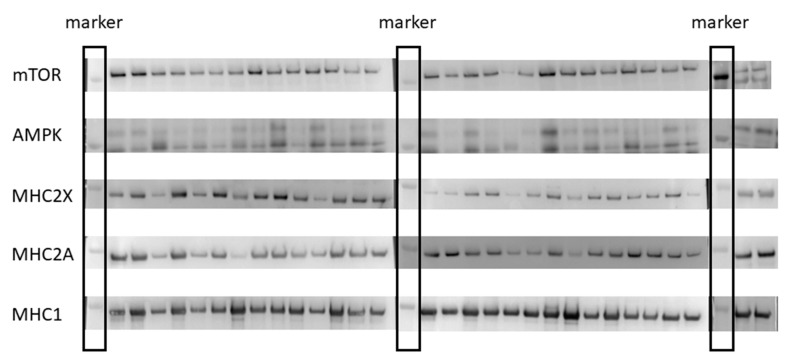
Immunoblots for mammalian target of rapamycin (mTOR), adenosine monophosphate kinase (AMPK), and the myosin heavy-chain (MHC) isoforms: MHC2X, MHC2A, and MHC1. Immunoblots were performed with a marker in the first lane, followed by the first subject’s pre-value and their post-value. This was repeated for all subjects and proteins.

**Figure 3 sports-07-00169-f003:**
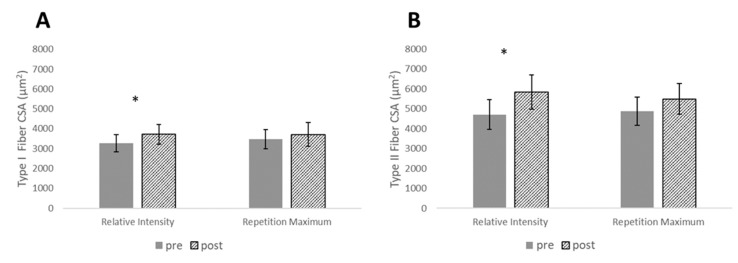
Changes in (**A**) type I and (**B**) type II cross-sectional area (CSA) pre- to post-intervention. * denotes significance for relative intensity group, *p* ≤ 0.05.

**Figure 4 sports-07-00169-f004:**
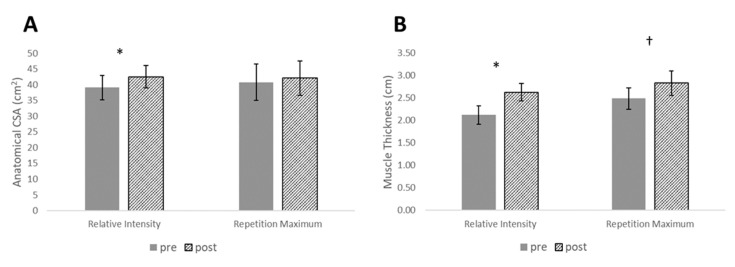
Changes in (**A**) anatomical cross-sectional area (ACSA) and (**B**) muscle thickness measured by ultrasonography pre- to post-intervention. * denotes significance for relative intensity group, *p* ≤ 0.05. † denotes significance for RM group, *p* ≤ 0.05.

**Figure 5 sports-07-00169-f005:**
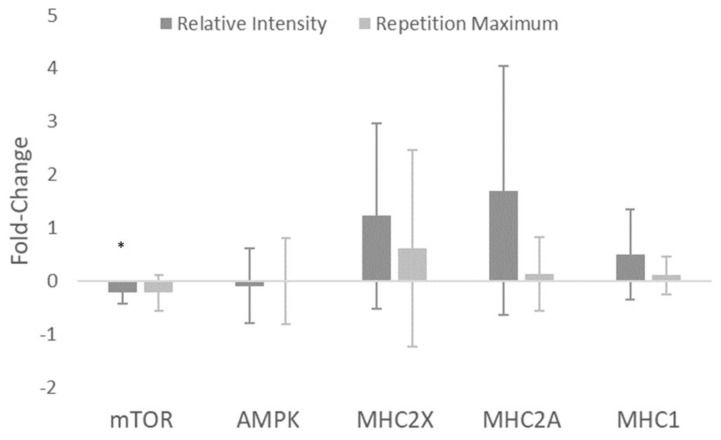
Fold-change results from immunoblotting for mammalian target of rapamycin (mTOR), adenosine monophosphate kinase (AMPK), and the myosin heavy-chain (MHC) isoforms: MHC2X, MHC2A, and MHC1. * denotes significance for relative intensity group, *p* ≤ 0.05.

**Table 1 sports-07-00169-t001:** Resistance Training Programs.

			RI_SR_		RM Zone
Training Block	Week	(Sets) × (Reps)	Day 1 and 2	Day 3	
(A) VJ and IMTP Testing					
Strength-Endurance	1	3 × 10	80.0%	70.0%	3 × 8−12
	2	3 × 10	85.0%	75.0%	3 × 8−12
	3	3 × 10	90.0%	80.0%	3 × 8−12
(B) VJ and IMTP Testing					
Max-Strength *	4	3 × 5	85.0%	70.0%	3 × 4−6
	5	3 × 5	87.5%	72.5%	3 × 4−6
	6	3 × 5	92.5%	75.0%	3 × 4−6
	7	3 × 5	80.0%	65.0%	3 × 4−6
(C) VJ and IMTP Testing					
Overreach	8	5 × 5	85.0%	75.0%	5 × 4−6
(D) VJ and IMTP Testing					
Speed-Strength	9	3 × 3	87.5%	67.5%	3 × 2−4
	10	3 × 2	85.0%	65.0%	3 × 1−3
(E) VJ and IMTP testing					

* Symbolizes down set at 60% of working weight (RI_SR_ only), RI_SR_ = relative intensity based on sets and repetitions, RM = repetition maximum, VJ = vertical jump, IMTP = isometric mid-thigh pull.

**Table 2 sports-07-00169-t002:** Training Exercises for all subjects.

Training Block	Day 1	Day 2	Day 3
Strength-Endurance	Back Squat, Overhead Press, Bench Press, DB Tricep Ext.	CG MTP, CG SLDL, BB Bent-Row, DB Bent Lateral Raise	Back Squat, Overhead Press, Bench Press, DB Tricep Ext.
Max-Strength	Back Squat, Push Press, Incline Bench Press, Wtd. Dips	CG MTP, Clean Pull, SG SLDL, Pull-Ups	Back Squat, Push Press, Incline Bench Press, Wtd. Dips
Overreach	Back Squat, Push Press, DB Step Up, Bench Press	CG CM Shrug, Clean Pull, CG SLDL, SA DB Bent-Row	Back Squat, Push Press, DB Step Up, Bench Press
Speed-Strength	Back Squat + Rocket Jump, Push Press, Bench Press + Med Ball Chest Pass	CG MTP, CG CM Shrug, Vertical Med Ball Toss	Back Squat + Rocket Jump, Push Press, Bench Press + Med Ball Chest Pass

DB = dumbbell, CG = clean grip, MTP = mid-thigh pull, BB = barbell, Ext = extension, Wtd. = weighted, SG = snatch grip, SLDL = stiff-legged deadlift, SA = single arm, CM = counter-movement.

**Table 3 sports-07-00169-t003:** Muscle size effect size using Hedge’s *g* and 90% confidence intervals for within-group and between-group effects.

	Relative Intensity Effects			Repetition Maximum Effects			Between-Group Effects
Variable	*g* (90% CI)	pre ± SD	post ± SD	*g* (90% CI)	pre ± SD	post ± SD	*g* (90% CI)
Type I CSA (µm^2^)	0.56 (0.22–0.89)	3277 ± 692	3720 ± 793	0.26 (−0.02–0.54)	3470 ± 789	3713 ± 974	0.48 (−0.35–1.31)
Type II CSA (µm^2^)	0.81 (0.37–1.26)	4079 ± 1195	5839 ± 1399	0.49 (−0.05–1.02)	4883 ± 1137	5493 ± 1241	0.50 (−0.33–1.33)
ACSA (cm^2^)	0.53 (0.33–0.73)	39.10 ± 6.25	42.53 ± 5.76	0.14 (0.00–0.28)	40.77 ± 9.22	42.09 ± 8.75	1.03 (0.20–1.86)
MT (cm^2^)	1.47 (0.99–1.95)	2.12 ± 0.33	3.62 ± 0.32	0.80 (0.46–1.14)	2.48 ± 0.38	2.83 ± 0.43	0.72 (−0.11–1.55)

*g* = Hedge’s *g* effect size, CI = 90% confidence interval, SD = standard deviation, CSA = cross-sectional area, ACSA = anatomical cross-sectional area, MT = muscle thickness.

**Table 4 sports-07-00169-t004:** Western blot effect size using Hedge’s *g* and 90% confidence intervals for within-group and between-group effects.

	Relative Intensity Effects		Repetition Maximum Effects		Between-Group Effects
	*g* (90% CI)	Fold Change ± SD	*g* (90% CI)	Fold Change ± SD	*g* (90% CI)
mTOR (AU)	−1.40 (−2.38–−0.43)	−0.22 ± 0.21	−0.97 (−1.86–−0.07)	−0.23 ± 0.33	0.02 (−0.80–0.85)
AMPK (AU)	−0.19 (−1.16–0.78)	−0.10 ± 0.70	−0.01 (−0.90–0.88)	−0.01 ± 0.81	−0.11 (−0.94–0.72)
MHC2X (AU)	0.93 (−0.04–1.90)	1.22 ± 1.74	0.44 (−0.46–1.33)	0.61 ± 1.85	0.31 (−0.52–1.14)
MHC2A (AU)	0.96 (−0.01–1.93)	1.70 ± 2.34	0.24 (−0.66–1.14)	0.13 ± 0.70	0.87 (0.02–1.73)
MHC1 (AU)	0.78 (−0.19–1.75)	0.50 ± 0.85	0.37 (−0.52–1.27)	0.10 ± 0.36	0.59 (−0.27–1.44)

*g* = Hedge’s *g* effect size, mTOR = mammalian target of rapamycin, AMPK = adenosine monophosphate protein kinase, MHC = myosin heavy chain.

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
