# Peer review of "Skeletal Muscle Fiber Adaptations Following Resistance Training Using Repetition Maximums or Relative Intensity"

_sports, 2019, doi:10.3390/sports7070169_

Round 1
Reviewer 1 Report
The manuscript is overall well written, the research question is relevant, the study design is appropriate, the results are clear and the Authors interpreted the results appropriately within the discussion. However, some questions remain to enhance the value of the manuscript. These are listed below.
Overall
· Please list the keywords
Abstract
· Please clarify what resistance training the two groups underwent (exercise and intensity).
· In the last sentence, please refer only to the present results.
Introduction
· Line 38: I believe “from the authors” not necessary.
· As a very first sentence, I’d expect RM and RIsr to be clearly defined.
· Line 44: Please clearly state the “performance outcome” you refer to and populate this first paragraph.
· The third paragraph does not clearly lead the reader towards the rationale of the study. This is mainly because of the reader has no methodological information about RM and RIsr. This is a very important point to be solved.
Methods
· The section “study design” and/or “procedures” is lacking.
· Resistance training: two main points:
1. Because the present rationale is all about the methods, please clearly describe the intervention the two groups underwent. I acknowledge these may have been published already, but this manuscript must contain them.
2. Equating the volume is not only about sets and repetitions. The Authors may refer to Coratella et al. 2016, Appl Physiol Nutr Met to show the exact volume calculation.
Discussion
· Lines 270-272: what the Authors proposed here is not supported by the introduction and methods. That’s why addressing the previous queries should be “mandatory”.
Author Response
Reviewer 1
The manuscript is overall well written, the research question is relevant, the study design is appropriate, the results are clear and the Authors interpreted the results appropriately within the discussion. However, some questions remain to enhance the value of the manuscript. These are listed below.
Thank you for your constructive comments.
Overall
· Please list the keywords
We believe the keywords are listed underneath the abstract.
Abstract
· Please clarify what resistance training the two groups underwent (exercise and intensity).
We have added a statement that, we believe, may address this comment.
· In the last sentence, please refer only to the present results.
Thank you, we have removed this statement from the abstract.
Introduction
· Line 38: I believe “from the authors” not necessary.
Thank you, we have made this change.
· As a very first sentence, I’d expect RM and RIsr to be clearly defined.
Thank you, we have made this change.
· Line 44: Please clearly state the “performance outcome” you refer to and populate this first paragraph.
We have added more specifics to this statement.
· The third paragraph does not clearly lead the reader towards the rationale of the study. This is mainly because of the reader has no methodological information about RM and RIsr. This is a very important point to be solved.
Our clarification in the first paragraph may have cleared up this issue, would you agree? If not, we can provide more information here as requested.
Methods
· The section “study design” and/or “procedures” is lacking.
· Resistance training: two main points:
1. Because the present rationale is all about the methods, please clearly describe the intervention the two groups underwent. I acknowledge these may have been published already, but this manuscript must contain them.
Thank you, we have added two tables which more clearly display the specifics of these training programs. If further clarification is required, please let us know.
2. Equating the volume is not only about sets and repetitions. The Authors may refer to Coratella et al. 2016, Appl Physiol Nutr Met to show the exact volume calculation.
Thank you for pointing this out. We acknowledge that training volume is not only about sets and repetitions. It would have been impractical for us to equalized time-under-tension in this investigation, therefore we aimed to equalize volume load only. To that end, we have amended the statement in our manuscript regarding this to state that we equalized volume load and not total training volume.
Discussion
· Lines 270-272: what the Authors proposed here is not supported by the introduction and methods. That’s why addressing the previous queries should be “mandatory”.
Thank you, we have added more detail regarding the intervention (Tables 1 and 2). We hope this will rectify this issue.
Reviewer 2 Report
This is a well conducted study which provides a well controlled and thorough analysis of appropriate and specific physiological responses to two forms of resistance training in well trained male subjects and acknowledges its main limitations. The interpretation of results is measured and appropriate. The importance of recovery in optimal adaptation is a critical emphasis.
One suggestion which could broaden its commentary would be for the authors to note that resistance training to failure has been thought to be critical in that it ensured maximal recruitment (and hence adaptation stimuli) of all type 2 muscle fibres. While the results indicate despite the lack of a regular training to failure component in the "relative intensity" training group, this group still demonstrated superior type 2 fiber adaptation. Further commentary by the authors on how optimal type 2 fibre recruitment and adaptation (beyond the important fatigue component) occurred in the "relative intensity" group would add to and further strenthen the discussion.
Author Response
Reviewer 2
This is a well conducted study which provides a well controlled and thorough analysis of appropriate and specific physiological responses to two forms of resistance training in well trained male subjects and acknowledges its main limitations. The interpretation of results is measured and appropriate. The importance of recovery in optimal adaptation is a critical emphasis.
Thank you very much for your positive and constructive comments!
One suggestion which could broaden its commentary would be for the authors to note that resistance training to failure has been thought to be critical in that it ensured maximal recruitment (and hence adaptation stimuli) of all type 2 muscle fibres. While the results indicate despite the lack of a regular training to failure component in the "relative intensity" training group, this group still demonstrated superior type 2 fiber adaptation. Further commentary by the authors on how optimal type 2 fibre recruitment and adaptation (beyond the important fatigue component) occurred in the "relative intensity" group would add to and further strenthen the discussion.
Thank you for this very insightful comment. We have added a few points of discussion regarding this.
Reviewer 3 Report
Overview
This is an excellent piece of work that examines the changes in skeletal muscle following a 'repetition max' or a 'relative intensity' training protocol. This has become a particular area of interest since some of the work, particularly from Canadian research groups, have shown similar levels of muscle protein synthesis with different iterations of training protocols than the typical 8-12 reps of 50-75% 1RM.
The paper is likely to be of broad interest within the field of Sports Science and I recommend it is accepted with some minor changes detailed below.
Minor comments:
Please could you include greater descriptions of the subject cohort such as age, weight, BMI, training backgrounds? This would be ideal if it can be summarised as a table with anthropometry of subjects
Somewhat related to above, you mention that only a small number had type IIx during staining; does this relate to training background of those individuals?
Why were statistics completed on two different software? Could all analyses not be completed on JASP?
Table 1 - the muscle CSA is sometimes reported in 1000's with a comma (e.g. 3,277) and sometimes without (e.g. 4709). Please make consistent throughout.
Line 220-221, and line 251-253 you report some statistical trends . Whilst I agree these are non-significant, do you not think the changes in MHC2X and MHC2A (for example) are simply because it the study is underpowered for these measures? I don't think this is a huge issue but it would be good to note this in the discussion. This way, future researchers that perhaps do find a statistically significant difference with (for example) n = 12/group know that this is not necessarily an opposing result
In figure 5 are these SEMs? What is the reason for the large variance? Did you have significant outliers? Have/could you exclude for 2 x SD's from the mean to see if you get the same results? I'm curious to see if these observations are driven by outliers...
Line 343 remove "
Congratulations on what is a very nice piece of work.
Author Response
Reviewer 3
Overview
This is an excellent piece of work that examines the changes in skeletal muscle following a 'repetition max' or a 'relative intensity' training protocol. This has become a particular area of interest since some of the work, particularly from Canadian research groups, have shown similar levels of muscle protein synthesis with different iterations of training protocols than the typical 8-12 reps of 50-75% 1RM.
Indeed! We thank the reviewer for their thoughtful comments on our manuscript.
The paper is likely to be of broad interest within the field of Sports Science and I recommend it is accepted with some minor changes detailed below.
Minor comments:
Please could you include greater descriptions of the subject cohort such as age, weight, BMI, training backgrounds? This would be ideal if it can be summarised as a table with anthropometry of subjects
Thank you, we have added age, body mass, and BMI (in addition to the baseline maximal strength values and resistance training experience). We have put this in the text, and we hope this is satisfactory.
Somewhat related to above, you mention that only a small number had type IIx during staining; does this relate to training background of those individuals?
We were not able to determine any connection with training history and IIx prevalence in our sample unfortunately. Perhaps if we are able to gather a larger sample of trained individuals we could tease this out in a future investigation.
Why were statistics completed on two different software? Could all analyses not be completed on JASP?
Unfortunately, many of the common statistical packages do not feature the use of Hedge’s g effect sizes, therefore we had to use Excel to calculate our effect sizes.
Table 1 - the muscle CSA is sometimes reported in 1000's with a comma (e.g. 3,277) and sometimes without (e.g. 4709). Please make consistent throughout.
Thank you, we have made this change.
Line 220-221, and line 251-253 you report some statistical trends . Whilst I agree these are non-significant, do you not think the changes in MHC2X and MHC2A (for example) are simply because it the study is underpowered for these measures? I don't think this is a huge issue but it would be good to note this in the discussion. This way, future researchers that perhaps do find a statistically significant difference with (for example) n = 12/group know that this is not necessarily an opposing result
Thank you for this comment. We have added a statement in the discussion regarding this.
In figure 5 are these SEMs? What is the reason for the large variance? Did you have significant outliers? Have/could you exclude for 2 x SD's from the mean to see if you get the same results? I'm curious to see if these observations are driven by outliers...
The error bars are confidence intervals in the figure. While we did not observe any statistical outliers, it is a fair observation that the variance was quite large. One possibility for this is we are measuring fold-changes in these proteins rather than being able to see actual concentrations/quantities. Similar to how %-change can be misleading at times, the fold-change scores can also result in the same misinterpretation. In order to rectify this, to the best of our ability, we have provided information beyond mean + SD. We have also included confidence limits for the effect sizes, in an effort to display the variance therein.
Line 343 remove "
Thank you, we have made this change.
Congratulations on what is a very nice piece of work.
Thank you again!
Round 2
Reviewer 1 Report
I commend the Authors for the improvements made on the manuscript.
There is no further query from this reviewer.